# Parameter Optimization and Fragmentation Prediction of Fan-Shaped Deep Hole Blasting in Sanxin Gold and Copper Mine

Bo Ke [1,2], Ruohan Pan [1], Jian Zhang [2,*], Wei Wang [1,3], Yong Hu [3], Gao Lei [3], Xiuwen Chi [1], Gaofeng Ren [1] and Yuhao You [1]

[1]  School of Resources and Environmental Engineering, Wuhan University of Technology, Wuhan 430070, China; boke@whut.edu.cn (B.K.); panrh@whut.edu.com (R.P.); 15871178782@139.com (W.W.); xwchi@whut.edu.cn (X.C.); rgfwhut@163.com (G.R.); a1622078050@gmail.com (Y.Y.)

[2]  School of Urban Construction, Wuchang University of Technology, Wuhan 430223, China

[3]  Hubei Sanxin Gold Copper Limited Company, Huangshi 435199, China; scb@hbsanxin.com (Y.H.); 13971757475@139.com (G.L.)

*  Correspondence: csuzhangjian@126.com

**Abstract:** For San-Xin gold and copper mine, deep blasting large block rate is high resulting in difficulty in transporting the ore out; secondary blasting not only increases blasting costs but is more likely to cause the top and bottom plate of the underground to become loose causing safety hazards. Based on the research background of Sanxin gold and copper mine, deep hole blasting parameters were determined by single-hole, variable-hole pitch, and oblique hole blasting tests, further using the inversion method to determine the optimal deep hole blasting parameters. Meanwhile, the PSO-BP neural network method was used to predict the block rate in deep hole blasting. The results of the study showed that the optimal minimum resistance line was 1.24–1.44 m, which was lower than 1.6–1.8 m in the original blasting design, which was one of the reasons for the higher blasting block rate. In addition, the PSO-BP deep hole blasting fragmentation prediction model predicts the block rate of the optimized blasting parameters and predicted a block rate of 6.83% after the optimization of hole network parameters. Its prediction accuracy is high, and the blasting parameter optimization can effectively reduce the block rate. It can reasonably reduce the rate of large pieces produced by blasting, improve blasting efficiency, and save blasting costs for enterprises. The result has wide applicability and can provide solutions for underground mines that also have problems with blasting large block rate.

**Keywords:** deep hole blasting parameters; blasting funnel; blasting fragmentation; neural network

## 1. Introduction

As a large industrial country, mineral resources play an important role in social infrastructure construction. However, due to over-exploitation, shallow mineral resources have been facing depletion, affected by mining equipment and mining technology. There are a certain number of hidden resources in the process of recovery. In order to meet the sustainable development of mines, mining enterprises often need secondary mining of these resources, and these hidden resources often present extremely complex conditions, especially in the filling body for recovery, in the surrounding filling body strength, poor stability of the quarry, difficulty in recovery, etc. How to ensure the stability of the filling quarry under the conditions of dynamic explosive load is a key technical problem facing the mining industry.

Blasting is the most important part of mine production. Its quality and efficiency are not only related to the production efficiency of the entire mining process, but also determine the economic benefits of the enterprise. In the blasting process, the control of

blasting fragmentation is very important. The fragmentation of blasting is an important indicator for evaluating the effect of blasting. Scholars have carried out a lot of research on the theory of fragmentation of rock blasting and have summarized a variety of control methods [1]. Shi Xiaopeng systematically studied the effect of blasting vibration on the filling body of large-diameter deep holes in Anqing copper mine using regression calculations and theoretical analysis, and then proposed technical measures to reduce the vibration intensity of large-diameter deep holes in Anqing copper mine based on the actual situation of blasting production [2]. In the case of Jinchuan's second mine, where the roof and the left and right helpers of the working face are filled bodies, Wang Xianlai made a series of elaborations on rock drilling, blasting material selection, hollowing method, number of holes, single shot dosage, and micro-difference time for blasting operations in the mining process, and concluded and optimized blasting parameters, which improved the safety of underground workers working under filled bodies and provided experience for reducing mine casualties [3]. However, the blasting process is very complicated, and the mining technology of different mines is different, resulting in a big difference between the prediction results of the fragmentation model and the actual ore fragmentation after blasting.

The degree of damage to the rock by explosives and the process control are mainly realized by blasting parameters. These parameters mainly include the nature of the explosives, the unit consumption of explosives, deep hole density factor, the minimum resistance line, the bottom distance of the hole, etc. These factors are in series with each other and determine the volume of the blasting funnel, which in turn affects the block size produced by blasting. Regarding the determination of blasting parameters, scholars have made a lot of explorations on the research of blasting parameters, optimizing blasting parameters from theoretical analysis, physical test (funnel test), engineering analogy, numerical simulation, and machine learning calculations. For example, Akande and Lawal examine optimization of blasting parameters for economic production of granite aggregates in Ratcon and NSCE quarries located at Ibadan using regression models [4]. Rezaeineshat et al. used robust techniques to design the blasting parameters in open-pit mine with the aim to reducing ground vibration [5]. Inanloo Arabi Shad and Ahangari proposed an empirical relation to calculate the proper burden in blast design of open-pit mines and promoted this method to other mines [6]. Wang et al. proposed a method combining UDEC and LS-DYNA to study the effects of parameters on the blasting effects [7]. Messaoud et al. classified the quality of rock mass through cluster analysis, and then studied the influence of different rock parameters on blasting through principal component analysis [8]. Monjezi and Dehghani used the neural network to analyze the relationship between the physical and mechanical properties of the rock mass, the performance of the explosive, the hole network parameters and the blasting backlash, optimize the blasting parameters, reduce the damage of the blasting backlash, and the optimization effect is obvious [9]. Sadollah et al. uses reverse neural network to optimize rock drilling and blasting parameters [10]. Similar research can be found in many other studies [11–13].

With the development of computer technology, in recent years, traditional theory combined with machine learning to predict the distribution of rock blasting fragmentation has gradually been recognized by everyone. Monjezi et al. established a 4-layer feed-forward artificial neural network (ANN) model to predict the distribution of fragmentation [14]. Ghiasi et al. predicted the large block rate generated in the open-pit blasting operation of Golegohar Iron Mine by using multiple regression methods and artificial neural network [15]. Ghaeini et al. and Ghaeini Hesarouieh et al. used the Mutual Information (MI) method to predict the blasting fragmentation of the Meydook Mine and compared it with the Kuz-Ram empirical model and statistical model. The results show that the MI model has higher accuracy than the Kuz-Ram and statistical models [16,17]. Based on meta-heuristics and machine learning algorithms, Xie et al. predicted the rock size distribution in mine blasting using various novel soft computing models [18]. Bahrami et al. implemented artificial neural network method to develop a model to predict rock fragmentation due to blasting in an iron ore mine [19]. Murlidhar et al. proposed a new hybrid imperialism

competitive algorithm (ICA)-artificial neural network (ANN) and found the proposed ICA-ANN model can be implemented better in improving the performance capacity of ANN model in estimating rock fragmentation during blasting [20]. Ouchterlony and Sanchidrián reviewed the prediction equations for blast fragmentation [21]. These models include some early fragmentation models, first Kuz-Ram models [22], crush zone model (CZM) and the two-component mode l(TCM) [23–25], extended Kuz-Ram model [26] and Swebrec function [27]. Some other methods were also adopted to predict rock fragmentation by many studies, for example, statistical modelling approach [28], muck-pile model [29], KCO model [30], and deep learning approach [31].

In fan-shaped deep hole blasting, the boulder yield is relatively high. How to take effective measures to reduce the output rate of boulders in deep hole blasting and improve the economic benefits of the mine is an urgent problem. Take Sanxin gold-copper mine as an example, the boulder yield is higher than 12%, in some stopes boulder yield even reached more than 20%. High boulder yield is not good for safe production and increases the workload of secondary blasting. This study takes Sanxin gold-copper mine as the research background and aims at reducing the boulder yield of this mine. Funnel tests [32] were conducted to determine the optimal blasting parameters including hole distance, explosive unit consumption, and row space. A combined particle swarm optimization (PSO) and BP neural network model was established to predict the boulder yield [33]. A targeted study was conducted to explore the effect of the maximum single-section charge on the filling body, and the safe distance of the filling body at different charges from the perspective of safe vibration speed, which provides a basis for the optimization of blasting parameters in the field.

## 2. Blasting Funnel Test

### 2.1. Test Equipment

Measuring tools include: weighing instrument (platform scale with weighing capacity $\geq$ 100 kg, electronic scale with sensing capacity $\leq$ 20 g), 50 m soft leather ruler, 5 m steel tape, shovel, gun stick and lattice screen (50 mm) $\times$ 50 mm, 100 mm $\times$ 100 mm, 200 mm $\times$ 200 mm, 300 mm $\times$ 300 mm, 400 mm $\times$ 400 mm grid).

### 2.2. Selection of Test Site

The selection of the blasting funnel test site should take into account that the ore lithology of the test orebody should be same or as close as possible to the mined area. According to the field investigation, the test site is selected at 8-3 rock drilling roadway at −609 m level in Sanxin Gold and Copper Mine. The funnel test site is shown in Figure 1.

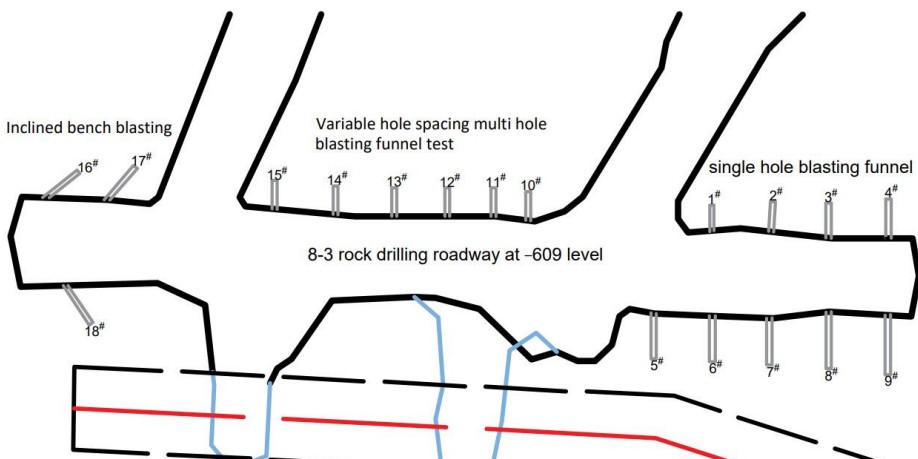

**Figure 1.** Plane layout of blast holes for series blasting funnel test. (In the figure, '#' is for the convenience of numbering the holes, 1# represents hole number one, and so on).

The physical and mechanical properties of the rock at the test site are shown in Table 1, and the performance parameters of the explosives used are shown in Table 2.

**Table 1.** Results of physical and mechanical properties of 609 horizontal ore and rock.

| Lithology | Bulk Density (N/m$^3$) | Tensile Strength (MPa) | Elastic Modulus (GPa) | Poisson's Ratio | Cohesion (MPa) | Internal Friction Angle (°) |
|---|---|---|---|---|---|---|
| Ore bearing marble | 32,300 | 6.01 | 41.007 | 0.24 | 17.88 | 50.5 |

**Table 2.** Main performance parameters of ANFO explosive.

| Explosive Type | Density (g/cm$^3$) | Explosion Velocity (m/s) | Stiffness (mm) | Explosive Force (ML) | Explosive Weight (g) |
|---|---|---|---|---|---|
| ANFO explosive | 1 | 3000 | 12 | 260 | 400 |

*2.3. Single Hole Blasting Funnel Test*

The designed blast hole depth is: there are 9 boreholes numbered 1–9 in 0.4 m, 0.5 m, 0.6 m, 0.7 m, 0.8 m, 0.9 m, 1.0 m, 1.1 m, and 1.2 m. Due to the errors in site construction and the sinking of the overlying rock layer caused by ground pressure, the actual depth of the blast hole is 0.4 m, 0.5 m, 0.6 m, 0.65 m, 0.7 m, 0.82 m, 0.9 m, 1.0 m, 1.1 m. The distance between holes is 2.4 m, the diameter of blast hole is 60 mm, and the charging length is 200 mm. Before charging, the individual deep hole is adjusted to the designed charge depth with gun mud, then the charge package is filled with gun stick, and finally the hole is plugged with 0.2 m gun mud. The ANFO explosive is initiated in reverse with the bottom of the detonator hole to reduce the blasting influence between the adjacent holes, and there are two non-adjacent holes in each blasting.

The layout of the blasthole is shown in Figure 2 below:

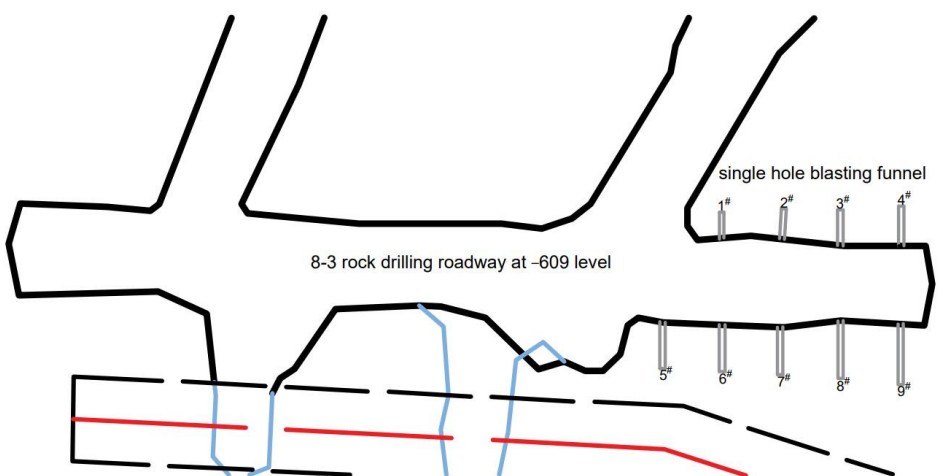

**Figure 2.** Schematic diagram of plane layout of blast holes in funnel test of single-hole blasting. (In the figure, '#' is for the convenience of numbering the holes, 1# represents hole number one, and so on).

*2.4. Variable Hole Spacing Multi Hole Blasting Funnel Test*

Six blastholes numbered 10–15 are arranged. The hole distance is 1.75, 2, 2.25, 2.5, and 2.75 times of the optimum funnel radius (the specific blasthole distance is 1.03 m, 1.18 m, 1.33 m, 1.48 m, 1.62 m), respectively. There are six holes in a group. The blasthole is arranged on the vertical free surface, and the hole diameter is 60 mm. The length of the charge is 200 mm. After blasting, the measuring process of single-hole blasting funnel test is repeated, and the funnel profile is drawn based on the paint on the grid.

The layout of the blasthole is shown in Figure 3 below:

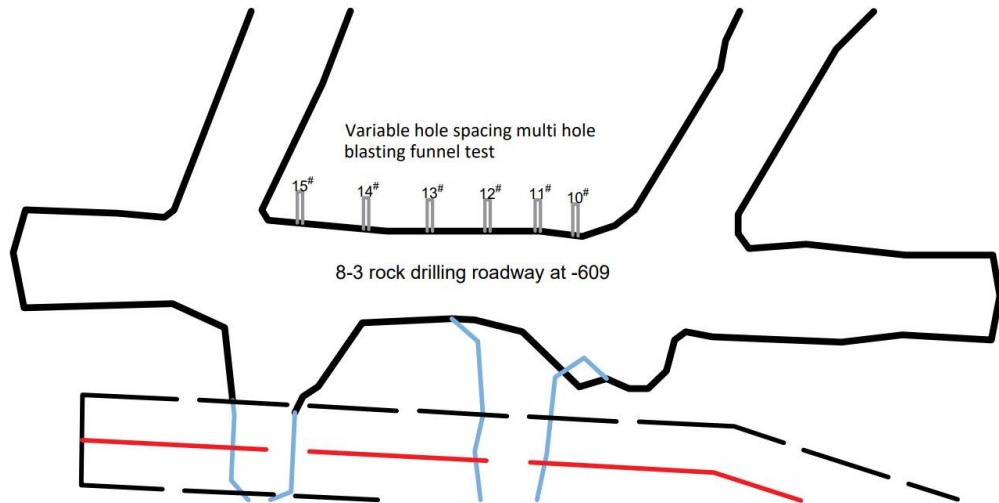

**Figure 3.** Schematic diagram of blast hole plane layout in funnel test of multi-hole blasting with variable hole distance. (In the figure, '#' is for the convenience of numbering the holes, 10# represents hole number one, and so on).

## 2.5. Inclined Bench Blasting Test

The design scheme is shown in Figure 4, and the blasthole layout is shown in Figure 5 below:

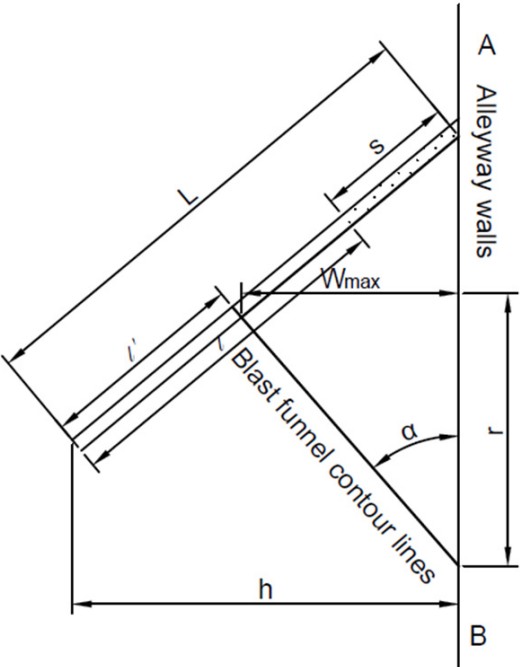

**Figure 4.** Schematic diagram of geometric parameters of inclined bench blasting.

In the drawing, AB is the roadway wall, L is the hole depth, m;
$l$—charge depth, m;
$l'$ is the residual blasthole depth, m;
$h$—the distance between the bottom of the blasthole and the roadway wall (free surface), m;
$r$—blasting funnel radius, m;

$s$—blasthole clogging depth, m;

$W_{max}$—maximum resistance line of blasting, m;

$\alpha$—is the angle between the blasthole and the free surface AB.

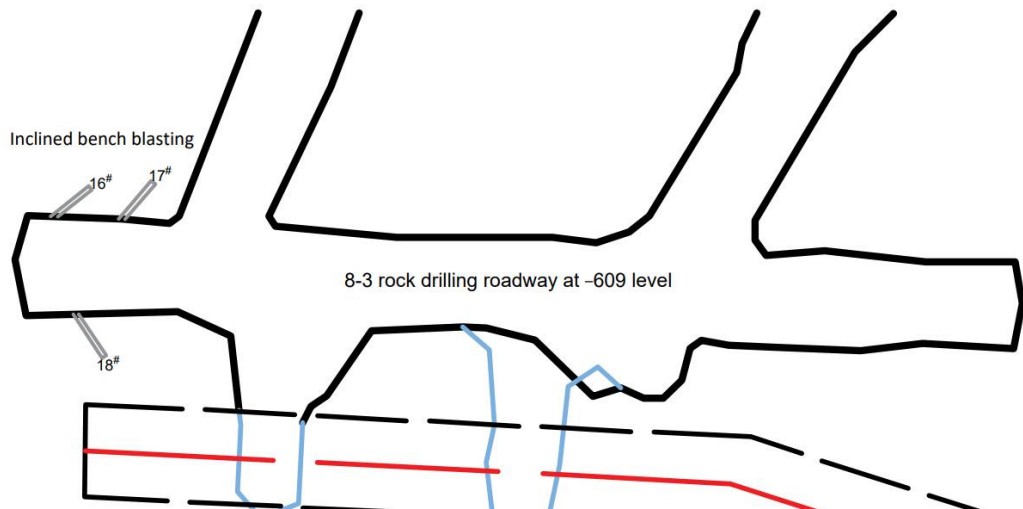

**Figure 5.** Schematic diagram of plane layout of hole in funnel test of inclined bench blasting. (In the figure, '#' is for the convenience of numbering the holes, 16# represents hole number one, and so on).

There are three blastholes in the inclined bench blasting experiment, numbered 16–18, drilling depth is the same, and the angles between them and the wall of the roadway are 30°, 40°, and 50°, respectively.

*2.6. Test Data Collection*

The measurements before and after blasting are as follows:

1.  Before blasting, the blasthole diameter, blasthole depth, explosive filling depth and packing length are measured respectively with steel tape measure, the explosive burying depth is recorded, and the colored stripe cloth is laid at the bottom of the roadway to ensure that the ore falling after blasting falls on the colored stripe cloth.
2.  After blasting, 200 mm × 200 mm grid is used to cover the blasting funnel surface, the grid is kept perpendicular to the roadway floor, the original free surface position was replaced with the grid plane, and the residual hole depth of the blasting funnel and the straight line distance between the bottom of the hole and the grid plane was measured.
3.  Deducting the falling part of the rock around the funnel mouth, taking the center line of the blasthole as the axis, the distance between the funnel boundary and the axis of eight different directions is directly measured every 45°, and the average value is taken as the blasting funnel radius.
4.  The grid is used to screen the ore after blasting, and the ores of different sizes after screening are weighed and recorded respectively.
5.  The rock fragmentation of blasting funnel test is screened. According to the actual production situation of the mine, the bulk above 400 mm is defined as unqualified. By weighing the ores of different sizes, the percentage of the total rock mass in blasting is calculated, and the blasting effect of blasting funnel test is compared.
6.  After inclined bench blasting, in order to ensure the accuracy of the measured data, a gun stick is inserted into the residual hole, and a tape measure is used to measure the resistance line at the opening formed by blasting, and the average value of the resistance line is the maximum resistance line $W_{max}$.

## 3. Data Processing and Blasting Parameter Inversion

### 3.1. Data Processing of Single-Hole Blasting Funnel

The hole 9 was blown up in the field blasting so that the data could not be measured; the single-hole blasting funnel test results were only obtained for single-hole blasting 1 to 8. The test results of single-hole blasting funnel are shown in Table 3.

**Table 3.** Funnel test results of single-hole blasting.

| Blasthole Number | Blasthole Depth (m) | Explosive Package Length (m) | Explosive Package Depth (m) | Funnel Volume (m$^3$) | Funnel Depth (m) | Funnel Radius (m) | Buried Depth Ratio | Unit Explosive Blasting Volume (m$^3$) |
|---|---|---|---|---|---|---|---|---|
| 1$^{\#}$ | 0.4 | 0.2 | 0.3 | 0.1305 | 0.27 | 0.371 | 0.29 | 0.3263 |
| 2$^{\#}$ | 0.5 | 0.2 | 0.39 | 0.1729 | 0.27 | 0.428 | 0.37 | 0.4322 |
| 3$^{\#}$ | 0.6 | 0.2 | 0.5 | 0.2166 | 0.33 | 0.606 | 0.48 | 0.5415 |
| 4$^{\#}$ | 0.65 | 0.2 | 0.55 | 0.2463 | 0.38 | 0.573 | 0.52 | 0.6157 |
| 5$^{\#}$ | 0.7 | 0.2 | 0.45 | 0.1963 | 0.3 | 0.523 | 0.43 | 0.4908 |
| 6$^{\#}$ | 0.82 | 0.2 | 0.72 | 0.1761 | 0.22 | 0.396 | 0.69 | 0.4402 |
| 7$^{\#}$ | 0.9 | 0.2 | 0.76 | 0.111 | 0.29 | 0.333 | 0.72 | 0.2775 |
| 8$^{\#}$ | 1 | 0.2 | 0.85 | 0.0916 | 0.16 | 0.298 | 0.81 | 0.229 |

The characteristic curve of the blasting funnel (V-L curve) and the curve of the relationship between the radius of the funnel and the burial depth of the explosive package (R-L curve) are generally characterized by polynomials, while the curve polynomials can be fitted based on the principle of least squares, and the accuracy of the test curve fit is mainly expressed by the sum of squares of the errors between the fitted and the test values of the polynomials. After the observation of the experimental data, the quartic polynomial is used to fit. MATLAB software was used to make the blasting funnel characteristic curve and the relationship curve between funnel radius and explosive package burial depth, as shown in Figures 6 and 7.

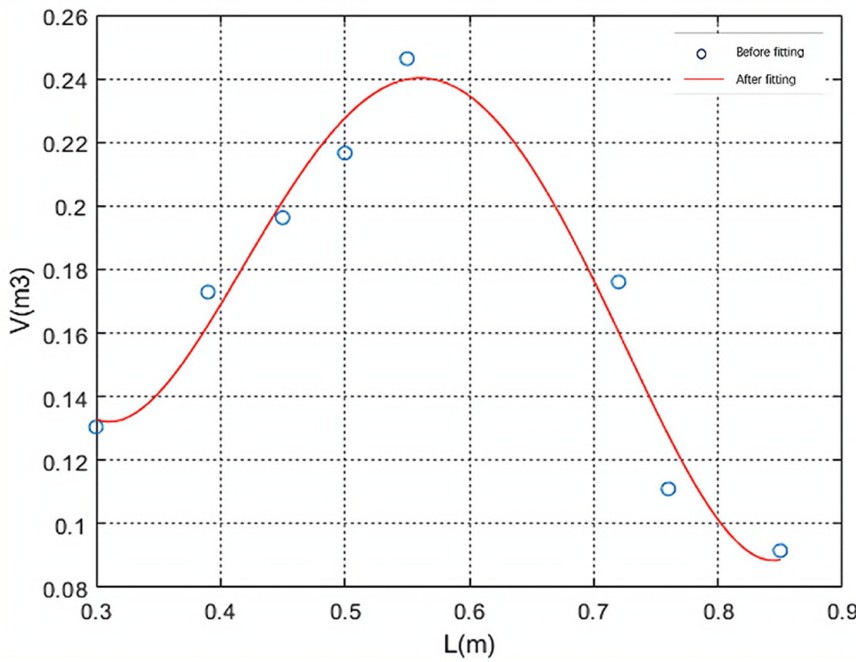

**Figure 6.** V-L blast funnel characteristic curve.

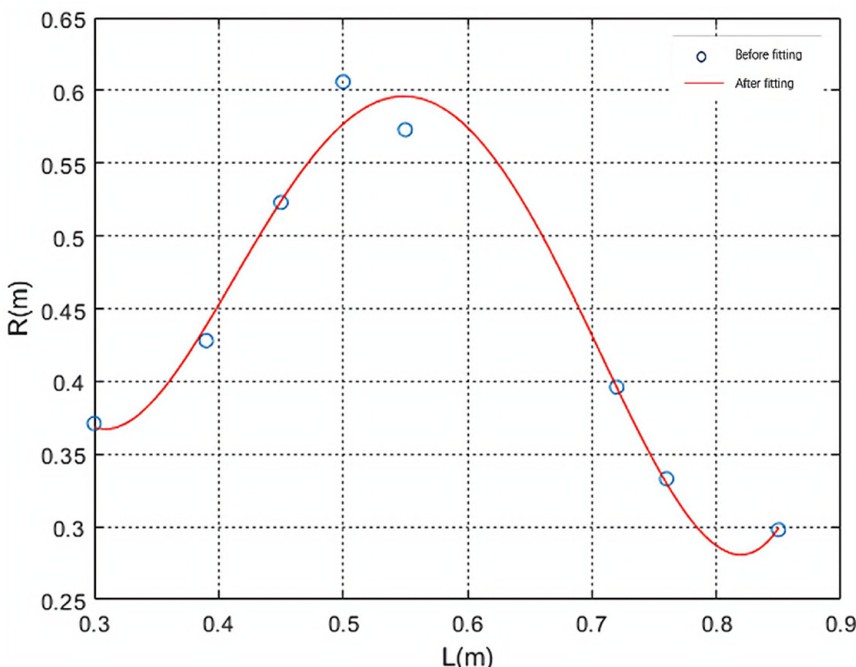

**Figure 7.** R-L funnel radius and explosive package burial depth relationship curve.

According to the principle of the least square method, the test data of the blasting funnel are regressed for four times, and the volume of the blasting funnel ($V_j$) and the buried depth of the explosive charge ($L_e$) are obtained. The multiple expressions between the radius of blasting funnel ($R_j$) and the buried depth of explosive charge ($L_j$) are as follows:

$$V = 25.44L^4 - 58.19L^3 + 46.29L^2 - 14.96L + 1.82 \tag{1}$$

$$R = 63.66L^4 - 142.36L^3 + 111.13L^2 - 35.41L + 4.32 \tag{2}$$

From Figures 7 and 8, we get $V_j = 0.24$ m$^3$ and $R_j = 0.59$ m. Substituting $V_j$ and $R_j$ into Equations (7) and (8) gives:

Optimum embedment depth: $L_j = 0.56$ m,

Critical buried depth of charge: $L_e = 0.57$ m

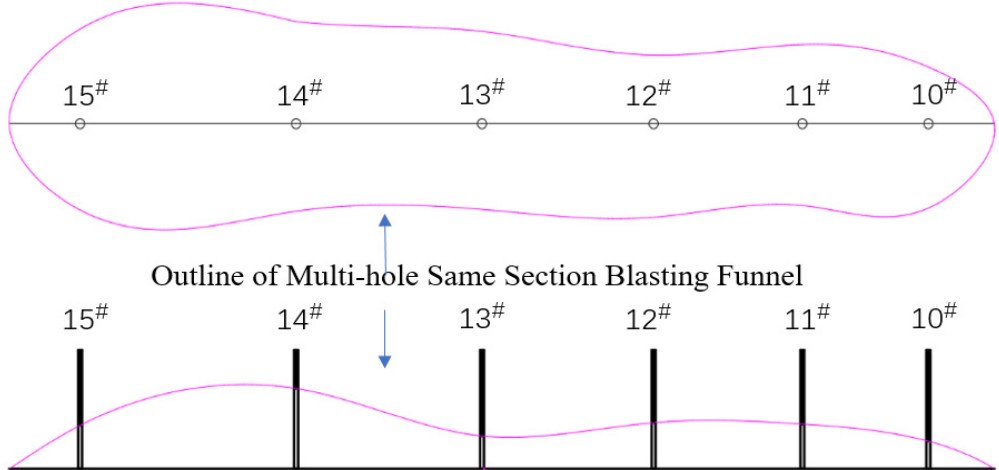

**Figure 8.** Outline of multi-hole same section blasting funnel.

### 3.2. Data Processing of Multi Hole Blasting Funnel with Variable Hole Spacing

See Figure 8 for the outline of blasting interface, and Table 4 for the test results:

**Table 4.** Test results of multi hole blasting funnel with variable hole spacing.

| Blast Hole Number | Medicine Bag Buried Depth/m | Blast Hole Spacing/m | Unit Explosive Blasting Volume (m³) | Volume of Blasting Funnel (m³) | Description of Blasting Fragmentation |
|---|---|---|---|---|---|
| 10~11[#] | 0.56 | 1.05 | 0.9483 | 0.3793 | Small and uniform lumpiness |
| 11~12[#] | 0.56 | 1.24 | 1.0478 | 0.4191 | Small and uniform lumpiness |
| 12~13[#] | 0.56 | 1.43 | 1.0601 | 0.4243 | Small and uniform lumpiness |
| 13~14[#] | 0.56 | 1.55 | 0.7503 | 0.3001 | The lumpiness is uniform, and there is basically no large block |
| 14~15[#] | 0.56 | 1.80 | 1.1315 | 0.4526 | Large fragmentation |

From the blasting effect observed in the field, it is known that a group of blastholes with a distance of 1.80 m are not connected into grooves after blasting, and basically form independent blasting funnels. After blasting, the four groups of blastholes with hole spacing of 1.05 m, 1.24 m, 1.43 m, and 1.55 m are connected to form grooves along the center line of the blasthole.

The above results show that when the hole spacing is small, the explosive blasting energy can be superimposed on each other, and the blasting effect is good, and there is basically no inter-hole spine. With the increase of hole spacing, the explosive blasting energy becomes weaker and the blasting effect becomes worse gradually. The spine between the holes gradually appears. When the hole spacing increases to a certain value, the line in the center of the hole no longer communicates to form a strip blasting groove, but develops into an independent blasting funnel in the triangular spine between the two holes.

According to the test results of porous blasting funnel in the same section, if the blasting parameters such as reasonable blasting hole spacing and explosive unit consumption are selected, the triangular spine between adjacent blasting funnels can be broken better. In order to ensure the blasting crushing effect, it is recommended that the hole spacing is 2.5 times of the optimum blasting funnel radius according to the blasting funnel test results, and hole spacing is 1.55 m. According to the above situation, under the test conditions, the optimum hole bottom distance is 1.43 to 1.55 m, and the blasting volume per unit explosive is 0.7503 m³.

*3.3. Data Processing of Inclined Bench Blasting Test*

In the experiment of inclined bench blasting, there are three blastholes, numbered 16[#], 17[#], and 18[#]. The destruction of the 16[#] blasthole during blasting, the data could not be measured. So only the data for blastholes 17[#] and 18[#] were obtained, and the angles between them and the wall of the roadway are 40° and 50°, respectively. Generally speaking, before blasting, the optimum burying depth obtained from the single-hole blasting funnel experiment is taken as the predicted median of the blasting funnel length. The hole length of the inclined bench blasting funnel experiment is 1.6 m and the explosive filling depth is 1.25 m. The layout and results of the test holes are shown in Table 5. The test results show that the average value of the best resistance line under the test conditions is 0.712 m.

**Table 5.** Test results of inclined bench blasting funnel.

| Blasthole Number | Hole Depth/m | Inclination Angle/° | Orifice Resistance Line/m | Hole Bottom Resistance Line/m | Charge Length/m | Best Resistance Line/m |
|---|---|---|---|---|---|---|
| 17[#] | 1.6 | 40 | 0.8 | 1.02 | 0.36 | 0.804 |
| 18[#] | 1.6 | 50 | 0.96 | 1.22 | 0.34 | 0.62 |
| average | 1.6 | 45 | 0.88 | 1.12 | 0.35 | 0.712 |

*3.4. Inversion of Stope Deep Hole Blasting Parameters*

On the basis of several blasting funnel tests in the field, the following blasting parameters are obtained when the blasthole diameter is 60 mm:

Optimum embedding depth: $L_j$ = 0.56 m;

Optimum funnel radius: $R_j$ = 0.59 m;

Critical buried depth of charge: $L_e$ = 0.57 m

Volume of blasting funnel at optimum burial depth: $V_j$ = 0.24 m$^3$;

Optimum hole bottom distance: $a$ = 1.43~1.55 m;

Weight of spherical charge: $Q_0$ = 0.4 kg;

According to Livingston's blasting funnel theory and based on the blasting energy balance criterion and similarity principle, many parameters suitable for deep-hole sector blasting in Sanxin Gold and Copper Mine need to be further deduced so as to provide scientific basis for the design of deep-hole sector blasting in the future.

The strain energy coefficient $E$ and the optimum burial depth ratio $\Delta_j$ can be determined by single-hole blast funnel tests:

$$E = \frac{L_e}{Q_0^{\frac{1}{3}}} \tag{3}$$

$$\Delta_j = \frac{L_j}{L_e} \tag{4}$$

where,

$L_e$—critical buried depth, m;

$E$—strain energy coefficient, which is constant for specific rocks and explosives;

$Q_0$—weight of spherical charge, kg;

$L_j$—optimum buried depth, m;

$\Delta_j$—optimum burial depth ratio. $\Delta_j$ is constant for specific rocks and explosives.

By substituting the Parameters (3) and the Equation (4), the strain energy coefficient $E$ = 0.77 and the optimum buried depth ratio $\Delta_j = 0.98$ are obtained.

3.4.1. Row Spacing

The row spacing (minimum resistance line) is an important basic parameter for the design of deep hole blasting, based on the principle that the amount of charge that can be loaded into a deep hole ($Q = \pi d^2 L \Delta / 4$) and the amount of charge required for a deep hole (unit volume of explosive consumption multiplied by the amount of blasting square that the hole is burdened with ($Q = WaLq$) are equal, obtaining the equation:

$$W = d\sqrt{\frac{0.785\Delta\tau}{mq}} \tag{5}$$

In the equation:

$d$—aperture, $d$ = 0.060 m:

$\Delta$—charge density, $\Delta$ = 1.0 × 10$^3$ kg/m$^3$;

$\tau$—deep hole charge coefficient, $\tau = \frac{\text{Charge length}}{\text{Gun hole length}} = 0.85$

$m$—deep hole density factor, $m = 1.0 \sim 1.2$

$q$—single consumption of explosives.

The effective explosive unit consumption approved by conventional blasting theory is:

$$q = \frac{Q}{f(n)W^3} \tag{6}$$

where: $f(n)$—exponential function of blasting action, $f(n) = 0.4 + 0.6n^3$,

$n$—blasting action index, $n = \frac{R}{L_j}$.

$Q$—weight of spherical charge, 0.4 kg;

Combining the two Equations (5) and (6) yields the following equation:

$$W = \frac{mQ}{0.785d^2 \Delta \tau f(n)} \tag{7}$$

Therefore, according to the above equation, the minimum resistance line $W = 1.24{\sim}1.44$ m can be determined.

It can be seen that the minimum resistance line of 1.6 to 1.8 m used in the original blasting parameters is too large, which may be one of the reasons for the high blasting block rate.

Vertical sector holes have a hole diameter of 75 mm, according to the slant hole blasting funnel test results and blasting cube root similarity law, when the blasting funnel test package in the best burial depth $W_1 = 0.712$ m, package radius $r_1 = 30$ mm. When the diameter of the blasthole is 75 mm (prototype), the radius of explosive charge is $r_2 = 37.5$ mm (prototype), and the blasting resistance line is $W_2$ (prototype). The Equation (13) should be satisfied:

$$\frac{W_1}{W_2} = \frac{r_1}{r_2} \tag{8}$$

Substitute the relevant data into the equation to obtain: b = $W_2 \approx 0.88$ m.

Therefore, the best minimum resistance line for the prototype spherical explosive package can be calculated by the spherical explosive package model test. According to other scholars' studies, the blasting effect of columnar charge in blastholes is different from that of spherical explosive packages. When the blasting effect is the same, the resistance line of the pillar charge package is greater than that of the spherical charge package if the amount of the charge is also the same.

$$W_z = K_w W_a \tag{9}$$

where: $W_z$—resistance line of cylindrical charge;

$W_a$—resistance line of spherical charge;

$K_w$—coefficient, $K_w = 1.2{\sim}1.6$, take $K_w = 1.25$

Substitute the relevant data into Equation (9) to obtain the minimum resistance line $W$ (row distance b) $W$ = b = 1.4 m.

### 3.4.2. Hole Bottom Distance

The bottom distance of the hole a and the minimum resistance line $W$ (or row distance b) of the fan-shaped gun hole satisfy the Equation (10).

$$a = m \times W \tag{10}$$

where: $m$ is the density coefficient, which is the ratio of the distance between the bottom of the hole and the minimum line of resistance.

As shown in Figure 9, $W < R$, $a > R$, the rock breaking area of a single hole is S $(a, W)$, and the two-hole blasting fragmentation zone just covers all the rock mass in the single-hole blasting area, and the superposition zone of the two-hole blasting fragmentation zone eliminates the area where the crushing force cannot be reached. It can be considered that all the rock masses in the single hole blasting area have been fully broken at this time, which is the ideal condition of rock mass fragmentation, and it is also the situation that the design wants to achieve. For this reason, the function S $(a, W)$ can be defined: $0 < W < R$, $0 < a < 2R$, make S obtain the maximum value as far as possible and the previously determined w to determine the optimal value of a. According to the geometric relationship in Figure 9, the explicit function expression of hole network area S $(a, W)$ can be obtained:

$$S = 2\left( \frac{\pi R^2}{2} - R^2 \arctan\frac{a}{2w} + \frac{aw}{4} \right) \tag{11}$$

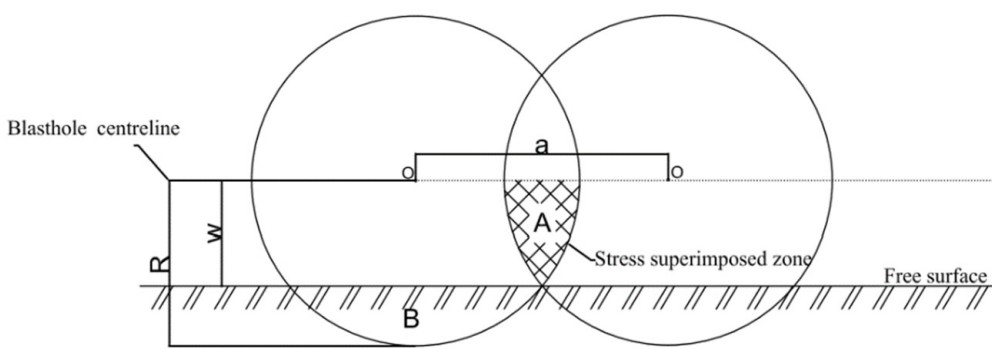

**Figure 9.** Stress superposition diagram at $0 < W < R$, $0 < a < 2R$.

Because: $\frac{1}{4}a^2 + W^2 = R^2$, in order to maximize the value of pore network area s, let the first derivative of s be zero, and according to the empirical calculation:

$$a = 1.23R \tag{12}$$

$$W = 0.789R \tag{13}$$

Then the deep hole density coefficient is: $m = a/W = 1.559$.

Based on the above equation calculation as well as the wide hole spacing multi-hole with section blasting funnel test, $m = 1.6$ is taken here, and small resistance line with large hole bottom distance blasting is implemented.

$$a = 1.6 \times 1.4 = 2.24 \approx 2.2 \text{ m} \tag{14}$$

3.4.3. Explosive Unit Consumption

From the single-hole and multi-hole with the same section blasting funnel test, it is known that when the explosive package is the best burial depth ratio and the blasting funnel volume is the largest, the explosive unit consumption is optimal. The best unit explosive consumption approved for single-hole and variable-pitch multi-hole tests by the optimal burial depth of the explosive package funnel blasting volume are $q_1 = 0.65 \text{ kg/m}^3$ ($q_1 = 0.4/0.6157 = 0.65 \text{ kg/m}^3$), $q_2 = 0.53 \text{ kg/m}^3$ ($q_2 = 0.4/0.7503 = 0.53 \text{ kg/m}^3$). However, the analysis shows that the test condition of explosive charge funnel blasting is the condition of full opening and free surface, and the cutting action of joints and fissures will increase the blasting ballast quantity to a certain extent, and the calculated explosive consumption index is low. At the same time, industrial production back blasting blocks up and down the disk for silica or quartz orthoclase diorite, the rock is loose and broken, its detonability is worse than the test lithology, and back blasting for extrusion blasting, and full free surface conditions are very different. Therefore, the test results must be corrected to ensure that the quality of recovery blasting and reduce the rate of large blocks.

Substitute relevant data into the Equation (6) to obtain: $q_3 = 0.706 \text{ kg/m}^3$.

The average explosive unit consumption obtained from the above three calculation methods is chosen as the average explosive unit consumption for the blasting design of the test section, which is taken as

$$q = \frac{q_1 + q_2 + q_3}{3} = 0.629 \text{ kg/m}^3 \tag{15}$$

**4. Prediction of Blasting Fragmentation Based on PSO-BP Neural Network Algorithm**

*4.1. PSO Optimized BP Neural Network Algorithm*

Due to the random selection of initial weights and thresholds during the application of BP networks, local convergence minima occur, which reduces the fitting effect, and to solve this problem, the initial weights and thresholds of the PSO optimized BP neural network

(PSO-BP) algorithm are used to solve the local minima problem and improve the prediction accuracy of the BP neural network algorithm. In PSO, $D$ is the dimension of the whole search space. See Equation (16) for the position of the $i$ particle.

$$X_i = [x_{i1}, x_{i2}, \cdots, x_{iD}]^T = \begin{bmatrix} \omega_{11} & \cdots & \omega_{nl} \\ \omega_{1m} & \cdots & \omega_{nm} \\ \theta_1{}^1 & \cdots & \theta_n{}^1 \\ \theta_1{}^2 & \cdots & \theta_n{}^2 \end{bmatrix}^T \tag{16}$$

$$D = (n+m)k + k + m \tag{17}$$

where: the elements of $\omega_{11} \cdots \omega_{nl} - \omega^1$
the elements of $\omega_{1m} \cdots \omega_{nm} - \omega^2$
the elements of $\theta_1{}^1 \cdots \theta_n{}^1 - \theta^1$
the elements of $\theta_1{}^2 \cdots \theta_n{}^2 - \theta^2$
$D$ is the dimension of this search space as the sum of all weights and thresholds in the BP neural network.

A population is randomly initialized with each particle setting a maximum velocity $V_{max}$ corresponding to an initial position maximum $X_{max}$ and each particle setting a minimum velocity $V_{min}$ corresponding to an initial position minimum $X_{min}$, and the positions and velocities of the particles are randomly selected in the interval $[-X_{max}, X_{max}]$, $[-V_{max}, V_{max}]$ for initialization, and each particle in the D-dimensional space in this population is represented by a set of weights and thresholds of the BP neural network. Set the termination condition of PSO optimization, the number of iterations N, the population size M, the initial value of inertia weight $\omega$, the initial value of acceleration factor $c_1$, $c_2$, and so on. According to the initial position of each particle, the fitness value $F(i)$ of each particle is obtained by the fitness function, see Equation (18).

$$F(i) = k(\sum_{i=1}^{n} abs(y_i - o_i)) \tag{18}$$

where: $n$—number of output nodes;
$y_i$—expected output of the $i$ node of the neural network;
$o_i$—predicted output of the $i$ node;
$k$—coefficient.
Set the extreme value $P_{gBest}$ of each particle population to the optimal value of the objective function; calculate the fitness value of each particle, if $F(i) > P_{iBest}$, $F(i)$ replaces $P_{iBest}$ and optimize the group extreme value $P_{gBest}$; update the speed and position of particles according to Equations (17) and (18) in each iteration; after the termination of PSO optimization, get the optimal weight and threshold of BP neural network; replace the BP neural network to initialize the initial weight and threshold of BP neural network, and input sample data. The BP neural network trains the model in the way of error reverse transmission. When the error between the output value of the BP neural network and the real value is large, the reverse error is calculated to reverse forward to update the connection weight and threshold between the layers until the termination condition is reached. The algorithm flow of PSO-BP is shown in Figure 10.

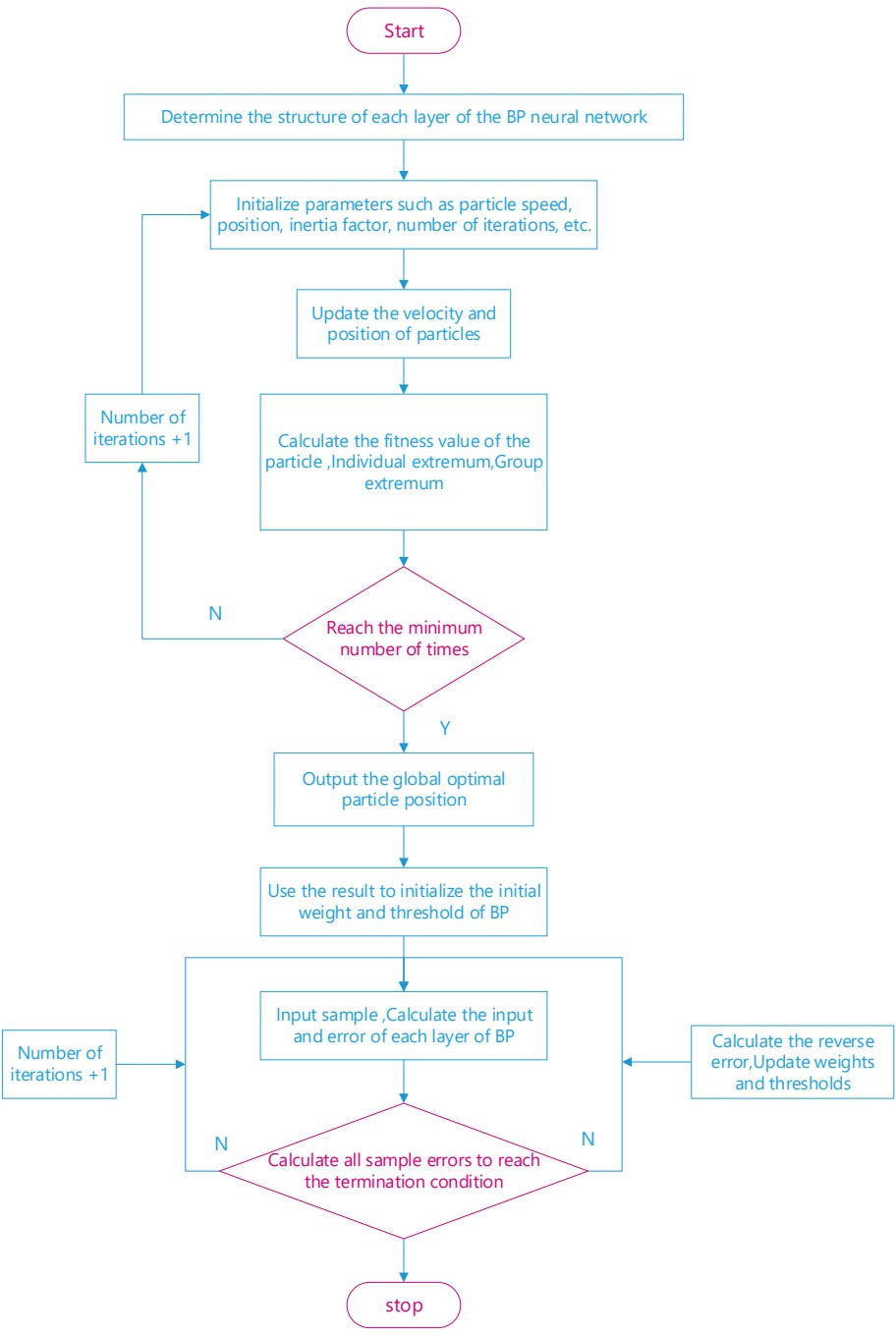

**Figure 10.** Algorithm flow of PSO-BP.

*4.2. Neural Network Model Structure*

There are many factors that affect the fragmentation distribution of blasting, and it is difficult to establish a mathematical expression to establish a relationship. How to highlight the principal contradiction from many influencing factors and select the main influencing factors to simplify the model as far as possible on the basis of ensuring the reliability of the data without reducing the prediction accuracy of the model is a problem that we should carefully consider before establishing the model.

1. There are many factors in the rock that can affect the blasting effect, and this model does not consider all the factors affecting the rock, but only selects some factors that have a significant impact on the blasting effect, such as the tensile strength of the rock, the compressive strength of the rock, and the spacing between the joints of the rock.

2. In terms of blasting technology and blasting parameters, combined with the existing equipment in the mine, this model only considers the simple blasting technology of regular hole layout, uniform charge and millisecond initiation with detonating cord, which is also the conventional production technology of the mine; in fact, the design of blasting technology is an art of flexible change.

3. There is no full consideration as to whether the charge is explosive coupled with blasting rock mass and whether the blasting energy can be fully utilized. At the same time, in the process of construction, many man-made uncertainties and equipment uncertainties cannot be fully considered.

4. The original data were normalized with $x' = 2 \times \frac{x - x_{\min}}{x_{\max} - x_{\min}} - 1$ because of different input factor magnitudes and large differences in order of magnitude, and the output data calculated by the network model are the normalized results, and the final data are obtained by reverse normalization for easy reading.

Determine the BP neural network structure of the deep hole blasting block degree prediction model; determine the initial position and velocity of the particle in space according to the BP neural network; set the termination condition of the particle population optimization, the number of iterations N, the population size, the initial value of inertia weight $\omega$, maximum particle speed $V_{max}$, the initial value of acceleration factor $c_1$, $c_2$, etc. According to Equation (18) calculate the particle's $F(i)$; according to $F(i)$ update $P_{iBest}$, $P_{gBest}$; according to Equations (1) and (2) update the particle position and velocity; calculate the new position of the particle $F(i)$; if $F(i) > P_{iBest}$, then $F(i)$ replaces $P_{iBest}$; optimize the population extremum $P_{gBest}$; in each iteration according to Equations (1) and (2) update the velocity and position of the particle; determine whether to reach the minimum error. If the error requirement is satisfied then PSO optimization is terminated, and after PSO optimization is terminated, the global optimal particle position is output, the optimal initial weights and thresholds of the deformation prediction BP network are obtained, and the training of the highway plane vertical deformation PSO-BP network is started.

Selection of input layer neuron parameters: rock characteristic parameters (density, compressive strength, joint spacing); seven blasting design parameters (explosive unit consumption, hole diameter, hole bottom distance, minimum resistance line) are used as the input layer of the neural network model. The data of the orthogonal experiments are shown in the following Table 6.

**Table 6.** Field orthogonal experimental data.

| No. | Density/(t·m⁻³) | Compressive Strength/Mpa | Joint Crack/ (Strip m⁻¹) | Aperture/mm | Hole Bottom Distance/m | Minimum Line of Resistance/m | Unit Consumption of Explosives/(kg·m⁻³) | Test Block Rate/% |
|---|---|---|---|---|---|---|---|---|
| 1# | 2.94 | 92.7 | 1.07 | 65 | 1.6 | 1.4 | 0.579 | 14.8 |
| 2# | 2.94 | 92.7 | 1.3 | 70 | 1.9 | 1.6 | 0.629 | 12.62 |
| 3# | 2.94 | 92.7 | 1.07 | 65 | 2.2 | 1.8 | 0.579 | 18.24 |
| 4# | 2.94 | 92.7 | 1.3 | 70 | 1.9 | 1.4 | 0.629 | 18.02 |
| 5# | 3.14 | 98.4 | 1.07 | 65 | 1.6 | 1.8 | 0.579 | 13.34 |
| 6# | 3.14 | 98.4 | 1.3 | 70 | 1.9 | 1.6 | 0.629 | 12.68 |
| 7# | 3.14 | 98.4 | 1.76 | 75 | 2.2 | 1.4 | 0.679 | 17.36 |
| 8# | 3.14 | 98.4 | 1.3 | 70 | 1.9 | 1.6 | 0.629 | 13.48 |
| 9# | 3.23 | 104.8 | 1.07 | 65 | 2.2 | 1.8 | 0.579 | 14.95 |
| 10# | 3.23 | 104.8 | 1.07 | 65 | 1.6 | 1.4 | 0.579 | 14.32 |
| 11# | 3.23 | 104.8 | 1.3 | 70 | 1.9 | 1.6 | 0.629 | 11.35 |
| 12# | 3.23 | 104.8 | 1.76 | 75 | 2.2 | 1.8 | 0.679 | 13.56 |
| 13# | 2.94 | 92.7 | 1.76 | 75 | 2.2 | 1.8 | 0.679 | 15.28 |
| 14# | 3.14 | 98.4 | 1.76 | 75 | 1.6 | 1.4 | 0.679 | 15.65 |
| 15# | 3.23 | 104.8 | 1.76 | 75 | 2.2 | 1.4 | 0.679 | 15.46 |
| 16# | 2.94 | 92.7 | 1.76 | 75 | 1.6 | 1.6 | 0.679 | 13.35 |
| 17# | 2.94 | 92.7 | 1.07 | 65 | 1.6 | 1.8 | 0.579 | 16.34 |
| 18# | 3.14 | 98.4 | 1.07 | 65 | 2.2 | 1.8 | 0.579 | 13.85 |
| 19# | 3.14 | 98.4 | 1.3 | 70 | 1.9 | 1.6 | 0.629 | 14.36 |
| 20# | 3.23 | 104.8 | 1.3 | 70 | 1.9 | 1.6 | 0.629 | 17.65 |
| 21# | 3.23 | 104.8 | 1.76 | 75 | 1.6 | 1.4 | 0.679 | 14.5 |

The data sets 1 to 12 in the orthogonal test table of the input data are used as the training sample set of the model, 13–15 as the validation set of the model and the data sets 16 to 21 and other 6 groups are also used as the test sample set. The experimental factor is used as the input factor of the network, and the result is used as the output factor.

### 4.2.1. Determine the Number of Hidden Layer Nodes

A three-layer neural network with infinite hidden layer nodes can realize the selection of the number of hidden layer nodes when constructing any neural network. If the number of hidden layer nodes is too small, the network cannot establish a complex mapping relationship, and the network prediction error is large. However, if there are too many nodes, the network learning time system will increase, and the phenomenon of "over-fitting" may occur, that is, the prediction of training samples is accurate, and the prediction error of other samples is larger. At present, there is no theoretical equation for calculating hidden layer neurons, but in the process of practical use, it is generally based on empirical Equation (19):

$$l < \sqrt{(m + n)} + a \tag{19}$$

where, $n$—number of nodes of input layer;

$l$—number of hidden layer nodes;

$m$—number of output layer nodes;

$a$—the constant of 0–10, where $n$ is 7 and $m$ is 1,

From Equation (19), the interval of the number of neurons in the hidden layer BP network can be determined as [2,13]. Analyze the effect of different number of neurons in the hidden layer on the prediction performance of the BP model. As can be seen from Figure 11, with the increasing number of hidden nodes, the standard deviation of the training error of the BP model gradually decreases, and the standard deviation of the simulation error also shows an overall decreasing trend; while the standard deviation of the average simulation error shows a trend of first decreasing and then increasing. This indicates that the network gradually transitions from the "under-learning" state to the "over-learning" state as the number of implicit neurons increases. When the number of hidden nodes is 10, the training standard deviation, the simulation standard deviation, and the average error all achieve the best results.

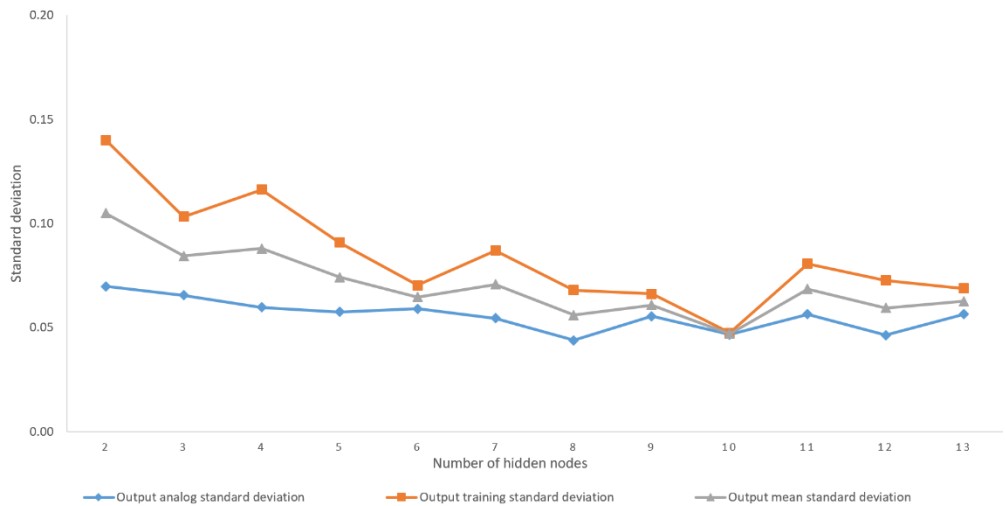

**Figure 11.** The effect of hidden layer nodes on the prediction effect.

### 4.2.2. Determining Inertia Weights $\omega$

It is shown [34] that the particle swarm algorithm performs best and has the highest probability of finding the global optimal solution when $0.8 < \omega < 1.2$. Take $\omega$ as 0.8, 0.9, 1.0, 1.1, 1.2, run ten times, and calculate the average value of the fitness, and the impact

results are shown in Figure 12. When $\omega = 0.9$, the global best fitness value calculated by the debugging parameters is the smallest and the model has the best solution performance.

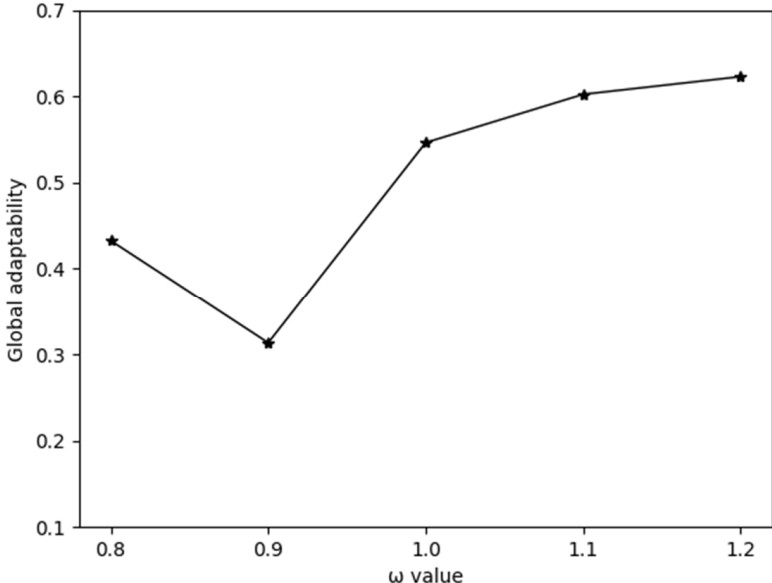

**Figure 12.** Effect of inertia weights on model performance.

### 4.2.3. Determining the Size of the Population

The population size M is generally taken from 20 to 90, and can be increased appropriately for complex or class-specific problems. However, the study shows that too large M values have no significant effect on improving the convergence accuracy of the model, while the computational complexity increases and the effect of finding the best becomes worse. M was divided into 15 groups and each group was run ten times to calculate the average value of the fitness, and the impact results were obtained as shown in Figure 13. When M = 70, the model had the best solution performance.

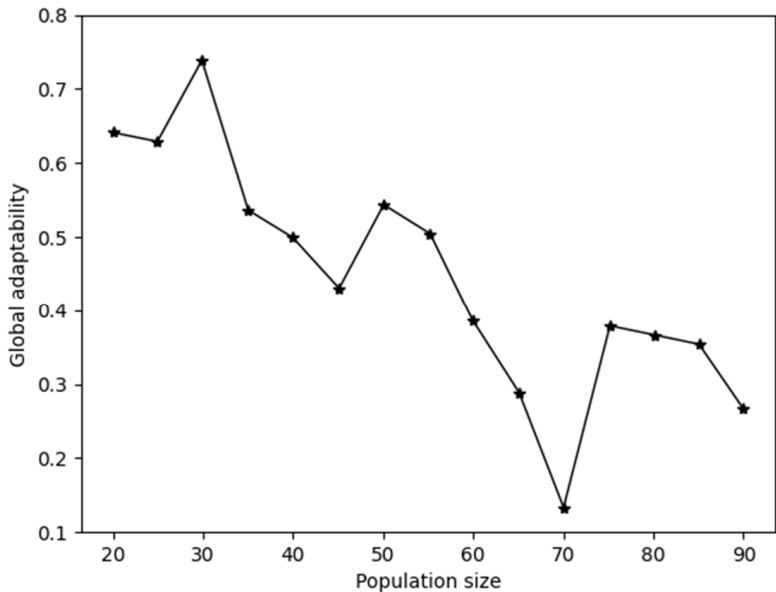

**Figure 13.** Effect of population size on model performance.

### 4.2.4. Determine the Maximum Velocity of the Particle

To speed up the model computation and prevent the model from diverging or falling into local optimum in the iteration, the maximum velocity $V_{max} = kX_{max}$ $(0.1 \leq k \leq 1)$ is

generally set, $X_{max}$ is the maximum search space, and $X_{max}$ is taken as 7. The $k$ values are divided into ten groups and run ten times to calculate the average value of the fitness, and the impact is derived as shown in Figure 14, when $k = 0.7$ and $V_{max} = 4.9$ the model has the best solution performance.

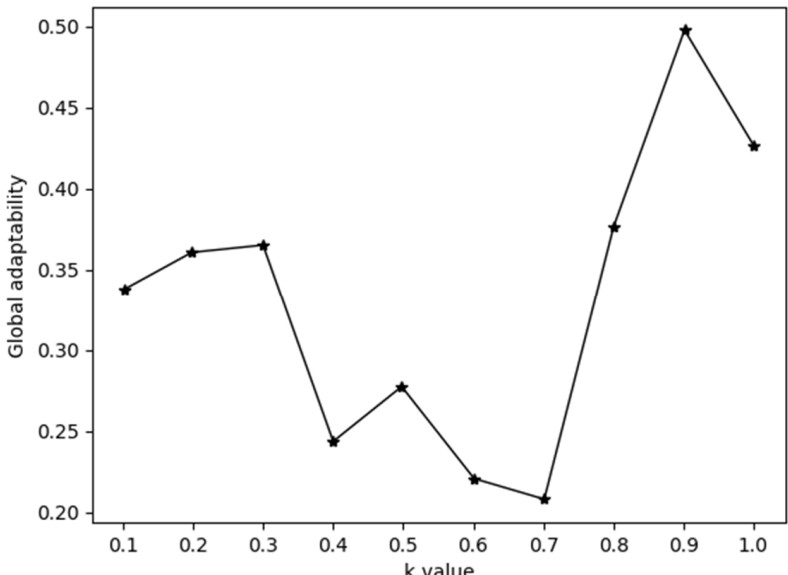

**Figure 14.** Effect of maximum particle velocity on model performance.

4.2.5. Number of Iterations

The PSO-BP algorithm is used to track the change of the fitness value of the objective function, and the change curve of the optimal individual fitness value of the particle swarm algorithm with the number of iterations is obtained, when the number of iterations reaches 15, the model gradually stabilizes, as shown in Figure 15.

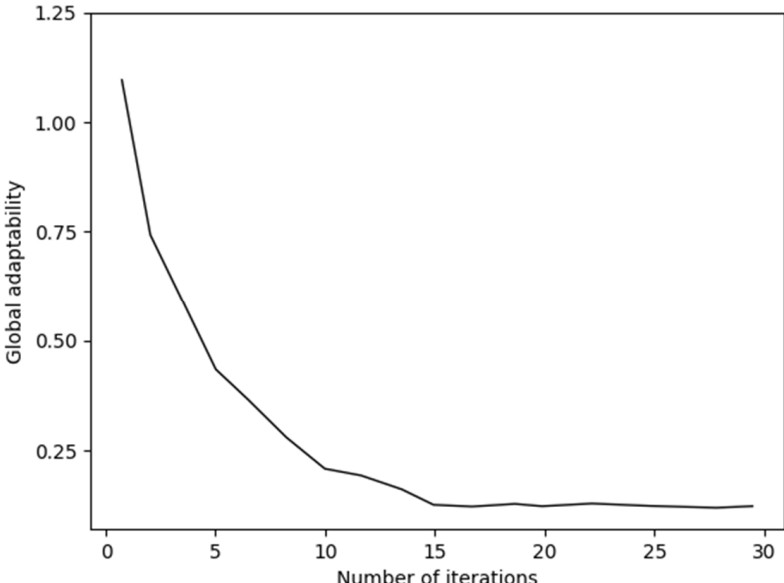

**Figure 15.** Effect of number of iterations on model performance.

The acceleration factors $c_1$, $c_2$ are dynamically adjusted with time [34] In summary, the model parameters are set as follows: number of nodes in the implicit layer l = 10, inertia

weight $\omega = 0.9$, population size $M = 70$, maximum velocity $V_{max} = 4.9$, $c_1, c_2 \in [0.5, 2.5]$. The structure of the PSO-BP neural network model is as Figure 16.

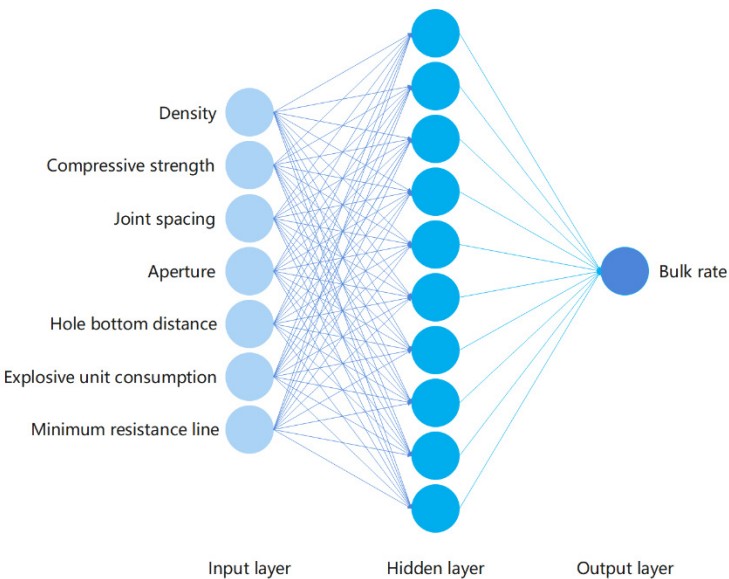

**Figure 16.** BP neural network model structure.

*4.3. PSO-BP Deep Hole Blasting Block Prediction*

The test sample Table 6 was input into the optimized PSO-BP neural network model, and the output values were compared as shown in the figure, trends almost overlap and the difference between the actual and predicted values was between $10^{-2}$ and $10^{-4}$.

The calculation results of error are shown in Figures 17 and 18. It can be seen from Figure 17 that the error between the real value and the predicted value obtained by PSO-BP neural network algorithm optimized by particle swarm optimization is very small. The high accuracy of parameters obtained by learning sample inversion shows that the algorithm is feasible and can be used to predict the fragmentation of deep hole blasting. As we can see from Figure 18, the test data one of the model regression plot has reached 0.88316. This shows the high accuracy of the prediction, which can be used for practical applications.

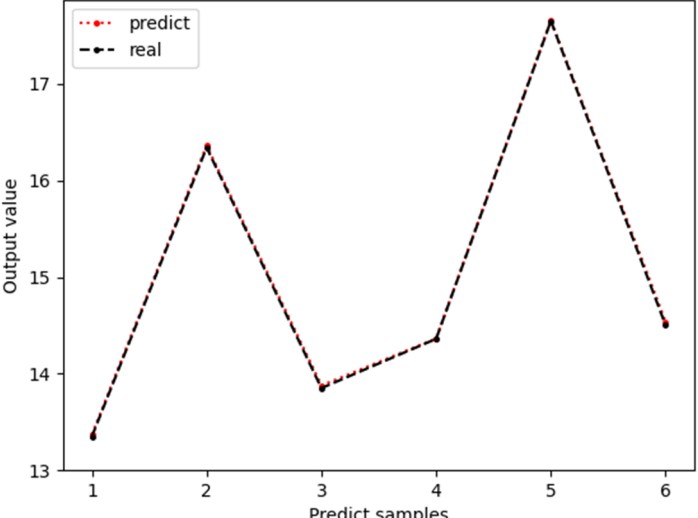

**Figure 17.** Comparison of actual and predicted values.

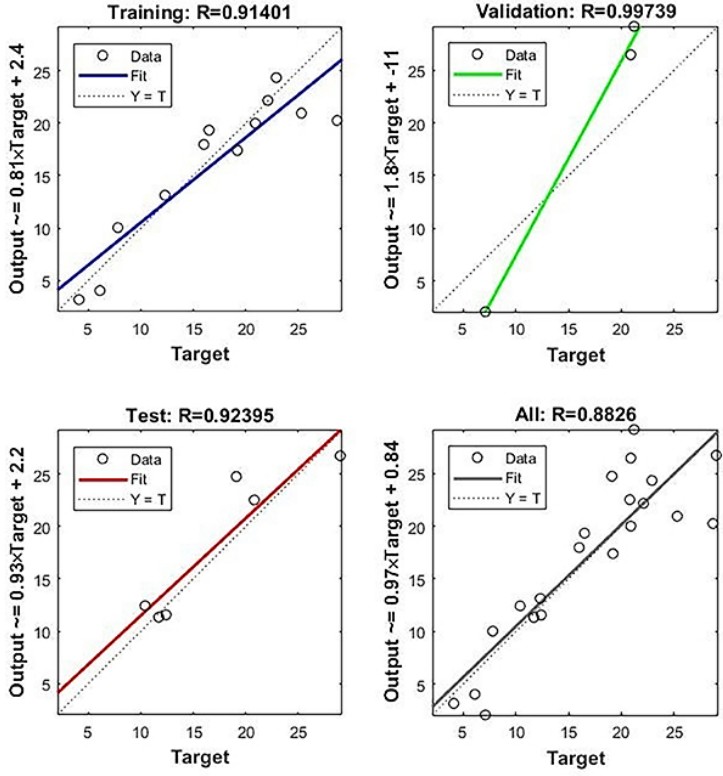

**Figure 18.** PSO-BP plot regression.

Further subdivision of hole bottom distance and minimum resistance line to predict blast block rate based on production accuracy requirements. Input the design samples for predicting the fragmentation of deep-hole blasting, and output the PSO-BP network prediction results of the fragmentation of deep-hole blasting after meeting the requirements of maximum iteration times or errors. Use PSO-BP network to calculate the data in Table 7. When the rock density is 3.14 t·m$^{-3}$, the compressive strength of the rock is 98.4 Mpa, and the Joint crack is 1.76 strip m$^{-1}$, the optimal blasting block rate of 6.83% can be achieved by arranging 75 mm drill holes, setting the bottom distance of the holes at 2.2 m, the minimum resistance line at 1.4 m, and the primary explosive unit consumption is taken as 0.629 kg·m$^{-3}$. This parameter has been applied to Sanxin Gold and Copper Mine Company.

**Table 7.** Predictive design data.

| No. | Density/(t·m$^{-3}$) | Compressive Strength/Mpa | Joint Crack/ (Strip m$^{-1}$) | Aperture/mm | Hole Bottom Distance/m | Minimum Line of Re-sistance/m | Unit Consumption of Explosives/(kg·m$^{-3}$) | Predicted Block Rate/% |
|---|---|---|---|---|---|---|---|---|
| 1# | 2.94 | 92.7 | 1.07 | 65 | 1.6 | 1.4 | 0.579 | 17.85 |
| 2# | 2.94 | 92.7 | 1.3 | 70 | 1.75 | 1.5 | 0.679 | 14.05 |
| 3# | 2.94 | 92.7 | 1.76 | 75 | 1.9 | 1.6 | 0.629 | 15.78 |
| 4# | 2.94 | 92.7 | 1.07 | 70 | 2.05 | 1.7 | 0.604 | 11.50 |
| 5# | 2.94 | 92.7 | 1.3 | 75 | 2.2 | 1.8 | 0.654 | 12.31 |
| 6# | 3.14 | 98.4 | 1.07 | 65 | 1.6 | 1.8 | 0.579 | 14.30 |
| 7# | 3.14 | 98.4 | 1.3 | 70 | 1.75 | 1.7 | 0.679 | 12.41 |
| 8# | 3.14 | 98.4 | 1.76 | 75 | 1.9 | 1.6 | 0.659 | 15.71 |
| 9# | 3.14 | 98.4 | 1.3 | 70 | 2.05 | 1.5 | 0.604 | 11.35 |
| 10# | 3.14 | 98.4 | 1.76 | 75 | 2.2 | 1.4 | 0.629 | 6.83 |
| 11# | 3.23 | 104.8 | 1.07 | 65 | 1.6 | 1.4 | 0.579 | 13.74 |
| 12# | 3.23 | 104.8 | 1.3 | 70 | 1.75 | 1.5 | 0.604 | 13.01 |
| 13# | 3.23 | 104.8 | 1.76 | 75 | 1.9 | 1.6 | 0.654 | 12.09 |
| 14# | 3.23 | 104.8 | 1.3 | 70 | 2.05 | 1.7 | 0.629 | 17.78 |
| 15# | 3.23 | 104.8 | 1.76 | 75 | 2.2 | 1.8 | 0.679 | 19.52 |

## 5. Conclusions

In this paper, the blasting parameters were optimized using blast funnel tests to address the problem of high blast block rate in the actual production of Sanxin gold and copper mine, and the blast parameters were optimized using PSO-BP neuron algorithm to predict the blast bulk output rate of sector deep hole blasting after the optimization of blast parameters, and the following conclusions were obtained in this study:

(1) Blast funnel tests were carried out in the 609 level to determine deep hole blasting parameters. In the field application of ammonium explosives, blasting funnel characteristics curve, blasting funnel volume and explosives burial depth relationship, funnel radius and explosives burial depth relationship, the best burial depth and critical burial depth of the blasting funnel, bursting pile block degree, and other content analysis and research were carried out, and then the deep hole blasting hole bottom distance, row spacing (minimum resistance line), and explosives unit consumption and other parameters were estimated. The final determination of the bottom distance of the hole is 2.2 m, the unit consumption of explosives is 0.629 kg/m$^3$, and the row spacing is 1.4 m.

(2) Establishment of PSO-BP deep hole blasting block prediction model, the rock properties and blast design parameters are selected as the input layer of the neural network, and the block rate as the output layer, the optimal number of hidden layer nodes is determined as 10, block prediction for deep sector blasting after optimization of hole network parameters, predicted block rate of 6.83% after optimization of orifice network parameters. It is predicted that its prediction accuracy is high and that blasting parameters can be optimized to effectively reduce the block rate.

(3) For underground mines with high blasting block yield, the results of the thesis can be applied with slight modifications based on the combination of the mine's own geological conditions, which can effectively reduce the generation of blasting block rate at the site, improving the mine's efficiency, and providing guidance to underground mining enterprises to reduce the blasting block rate in blasting construction.

**Author Contributions:** Conceptualization, B.K.; methodology, B.K.; software, Y.Y.; validation, R.P. and Y.Y.; formal analysis, B.K. and X.C.; investigation, R.P. and Y.Y.; resources, Y.H., W.W. and G.L.; data curation, R.P.; writing—original draft preparation, Y.Y. and R.P.; writing—review and editing, B.K. and J.Z.; visualization, Y.Y.; supervision, X.C. and G.R.; project administration, X.C.; funding acquisition, X.C. All authors have read and agreed to the published version of the manuscript.

**Funding:** This study was supported by the Fundamental Research Funds for the Central Universities (WUT: 2021IVA040).

**Data Availability Statement:** Not applicable.

**Conflicts of Interest:** The authors declare no conflict of interest.

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
