# Peer review of "Parameter Optimization and Fragmentation Prediction of Fan-Shaped Deep Hole Blasting in Sanxin Gold and Copper Mine"

_minerals, doi:10.3390/min12070788_

Round 1

Reviewer 1 Report

This manuscript can be employed after modification. The modification questions are as follows:
1. In the abstract and its introduction, the specific scientific problems to be studied are not clearly revealed.
2. Figures 1, 2, 3 and 5 are rough, and there is too little information related to this study.
3. In Table 5, pay attention to the writing standard of international units, such as "m" is not "m"
4. Formulas (3, 4, 5, etc.) in this paper are lack of references, and some formulas are not visible, such as "formula" under formula 10 and formula 18
5. The unit expression of explosive unit consumption is not uniform. Please check carefully, such as the calculation formula of formulas 11 and 12.
6. In this paper, there are too many experiments and neural network algorithms, which need to be simplified.
7. It is suggested to supplement the analysis of the influencing factors of blasting fragmentation, such as what are the influencing factors of blasting, how are the influencing laws, and which key factors need to be analyzed and revealed.
8. The conclusion needs to be concise, and the academic conclusion should better reflect the original characteristics and reflect the general law.

Reviewer 2 Report

Developing the use of machine learning methods is an important step for automation of engineering activities with heterogeneous operating conditions such as blasting in mining engineering. This research work has been done in this regard and can be considered for publication after minor modifications.

The article's English language needs to be reviewed.

For example:

Some sections are incomprehensible or hard to follow.

“Based on the research background of Sanxin Gold and Copper Mine, the parameters of deep-hole blasting are determined by single-hole blasting funnel test, variable hole spacing multi-hole blasting funnel test and inclined bench blasting test, further use the inversion method to determine the optimal deep hole blasting parameters.”

The results of the study showed that the optimal minimum resistance line was 1.14-1.24m, which was lower than the 1.6-1.8m in the original blasting design, which was one of the reasons for the higher blasting bulk rate.

It is better to be corrected with shorter sentences.

Please refer to new articles in this journal (Minerals) about blasting fragmentation optimization.

Minerals | Free Full-Text | An Evaluation on the Impact of Ore Fragmented by Blasting on Mining Performance (mdpi.com)

Please number pages.

Please correct the problems with reference. Example: Page 17 equation 18; Page 14 equation 17.

Author Response

Point 1: The article's English language needs to be reviewed.

Response 1: Thank you for your review, I have modified it according to your suggestion.

Point 2: Please number pages.

Response 2: Thank you for your suggestion, I have added the page number.

Reviewer 3 Report

Main comments:

1) p. 9 in Order to Ensure The Blasting Crushing Effect, It is Recommeded that Hole Spacing Is 2.5 Times of the Optimum Blastius According to the Blasting Funnel Test Stest Results. Who is recommended? Make a link

2) paragraph 3.4.2 - error in the formula Q2 =...

3) p. 18. "According to the ABOVE EMPIRICAL EQUATION, Throuch Continous Attempts, IT Was Finally Determined that Network Prediction Effect Satiscafactory When THEBER OF HIDDENENNE LAY The results are not given for others Number of Hidden Layer Nodes

4) Why in figure 11 3 hidden nodes?

5) It is necessary to present the result of a test of data sufficiency to create a neural network.

6) Fragmentation results are not presented

Reviewer 4 Report

  1. in the abstract, the optimal minimum resistance line was 1.14-1.24m. In table 11, the minimum line of resistance is 1.4m. Can author explain the reason?
  2. In section 2.4, the hole distance is 1.75,2,2.25,2.5 and 2.75 times of the optimum funnel radius (R=0.59). The hole distance of 1.05m,1.24m,1.43m,1.55m,1.80m seems to be not correct.
  3. The blasting parameters listed in Table 3 need to be revised in a more readable form.

4, What equation is used to calculate the Critical buried depth of charge (Le)?

  1. In section 3.3, the suggested spacing is 2.5times of funnel radius (1.55m), the blasting volume per unit explosive should be 0.7503 m3/kg.
  2. What is value of Q0?

7.In equation (9), a=1,6 x 1.3, can author explain the value 1.3?

  1. In section 3.4.2, where are the q1 and q2 values come from?
  2. The equation (10) need to be presented more clearly.
  3. Can authors explain equation (12) Δ=1.0*103 kg/m3?
  4. The authors need to make the statement that the values listed in Table 6 and 7 are design value but not measured value.
  5. The block rate, chunk rate and bulk rate mentioned in the manuscript are the same meaning?
  6. Table 8 is not clearly presented.

Reviewer 5 Report

The paper called Parameter optimization and fragmentation prediction of fanshaped deep hole blasting in Sanxin gold and copper mine by Bo Ke, Ruohan Pan, Jian Zhang, Wei Wang, Yong Hu ,Gao Lei, Xiuwen Chi ,Gaofeng Ren and Yuhao You is quiet good. There are some major aspects I would like to highlight.

  • The publication is very well described in terms of theoretical.
  • The paper should be prepared in accordance with the mpdi standard,
  • The purpose and scope of the research is incomprehensible, please correct it.
  • How the results obtained can be used in the future, please answer in the conclusions
  • In the presented summary, the importance of the publication should be briefly described, including the broader consideration of the methods, analyzes and results of the obtained research.
  • It should be emphasized what is the main scope and purpose of the presented publication, it's hard to find in the text.
  • What contribution does this work make to science?

The presented conclusions may be of fundamental importance, therefore they should be presented in a better light and the author’s should emphasize the original research contribution. I believe, that suggested amendments will significantly increase the relevance of the publication and will improve it. After applying all required changes, the paper is suitable for publication.

Round 2

Reviewer 3 Report

The article has really improved a lot, but there are still comments:

1) I didn't see the change in Point 4 ( Why in figure 11 3 hidden nodes?)

2) In fact, the training sample is represented not by 16 data sets, but by 12, since data 3 and 4 are identical, and data 13,14,15,16 are also identical.

3) The test sample also contains not 5 data sets, but 3.

4) Not shown for each dataset is the block rate value that is used as the output layer.

5) I did not see the results of the check on Point 5 ( It is necessary to present the result of a test of data sufficiency to create a neural network)

Reviewer 5 Report

Thank you for the changes made.  Accept in present form

Author Response

Thank you for your suggestion, I have modified it according to your request